

# The effects of isometric hand grip force on wrist kinematics and forearm muscle activity during radial and ulnar wrist joint perturbations

Kailynn Mannella[1], Garrick N. Forman[1], Maddalena Mugnosso[2], Jacopo Zenzeri[2] and Michael W. R. Holmes[1]

[1] Department of Kinesiology, Brock University, St. Catharines, Canada
[2] Robotics, Brain and Cognitive Sciences, Istituto Italiano di Tecnologia, Genoa, Italy

## ABSTRACT

The purpose of this work was to investigate forearm muscle activity and wrist angular displacement during radial and ulnar wrist perturbations across various isometric hand grip demands. Surface electromyography (EMG) was recorded from eight muscles of the upper extremity. A robotic device delivered perturbations to the hand in the radial and ulnar directions across four pre-perturbation grip magnitudes. Angular displacement and time to peak displacement following perturbations were evaluated. Muscle activity was evaluated pre- and post-perturbation. Results showed an inverse relationship between grip force and angular displacement ($p \leq 0.001$). Time to peak displacement decreased as grip force increased ($p \leq 0.001$). There was an increase in muscle activity with higher grip forces across all muscles both pre- and post-perturbation ($p \leq 0.001$) and a greater average muscle activity in ulnar as compared to radial deviation ($p = 0.02$). This work contributes to the wrist joint stiffness literature by relating wrist angular displacement to grip demands during novel radial/ulnar perturbations and provides insight into neuromuscular control strategies.

## INTRODUCTION

Optimal function and control of the distal upper extremity is critical for interacting with objects and performing nearly all tasks of daily life. As the end effector of the upper extremity, the wrist and hand can perform both forceful exertions, as well as precise movements to effectively execute fine motor tasks. An abundance of literature exists investigating neuromuscular control of the wrist joint (*Forman et al., 2020a*; *2020b*; *Charles & Hogan, 2010*; *2011*) and wrist joint stiffness during flexion and extension movements (*Charles & Hogan, 2012*; *Milner & Cloutier, 1993*; *De Serres & Milner, 1991*). *Charles & Hogan (2011)* demonstrated that wrist rotation dynamics are vastly different from arm reaching movements, since wrist rotations are influenced more by stiffness than inertia. The authors concluded that a primary aspect of wrist joint control involves compensating for stiffness. Thus, knowledge of wrist joint stiffness and the viscoelastic characteristics that

Corresponding author
Michael W. R. Holmes,
mholmes2@brocku.ca

govern stiffness is essential to understanding how humans perform coordinated wrist actions (*Formica et al., 2012*; *Drake & Charles, 2014*). However, most wrist and hand actions include concurrent wrist exertions and gripping tasks (*Forman et al., 2019*; *Forman, Forman & Holmes, 2021*) and the viscoelastic characteristics of the mechanical impedance of the wrist joint can change with gripping tasks (*Falzarano et al., 2021*). Our early work investigated muscular contributions to wrist joint rotational stiffness in flexion and extension (*Holmes, Tat & Keir, 2015*). We implemented a paradigm by which increases in forearm muscle co-contraction (*via* gripping) manipulated wrist stiffness, such that muscular responses to sudden loading could be investigated with tasks representing muscular demands of tool/object manipulation. To date, little is known about muscular responses to sudden loading in radial and ulnar deviation. *Forman et al. (2020c)* investigated muscular responses to radial/ulnar perturbations during a dynamic wrist flexion/extension task and passive wrist stiffness has only been evaluated across radial/ulnar directions in a small sample with a clinical focus (*Rijnveld & Krebs, 2007*). Thus, the aim of this work was to quantify how grip force demands influence wrist kinematics and forearm muscular contributions to sudden radial/ulnar perturbations.

Perturbations, due to sudden (often unexpected) external forces, are commonplace in many occupations, particularly when working in an environment where external interactions may occur between worker and surroundings. Understanding the effect of perturbations on task performance is critical to our basic understanding of neuromuscular control and may provide future insight into workplace safety. Anticipatory muscle activity and joint stiffening is one method commonly used in the presence of instability or perturbing forces, particularly when a perturbation is expected, to help minimize joint displacement caused by a perturbing force (*Forman et al., 2020a*; *Holmes, Tat & Keir, 2015*). Increasing grip force has been shown to increase forearm muscle activity (*Mogk & Keir, 2003*) and co-contraction (*Holmes, Tat & Keir, 2015*), which is considered one determinant of wrist stiffness. However, pre-emptive muscle activation is metabolically wasteful (*Hogan, 1984*) and may lead to an earlier onset of muscle fatigue.
The development of muscle fatigue will subsequently decrease maximal force production capabilities, reduce one's ability to effectively perform accurate fine motor tasks (*Forman et al., 2020b*; *Huysmans et al., 2008*), and can ultimately increase the risk of developing MSDs (*Dugan & Frontera, 2000*). Sufficient joint stiffness is needed to protect the joint while also allowing for accurate, unhindered movement of the wrist to complete a desired task. Compensatory actions post-perturbation *via* voluntary and involutory muscle responses will also influence task performance and injury risk and should be considered to completely understand neuromuscular control during perturbations. Perturbation magnitude, direction, timing, and background muscle activity can all influence compensatory actions at the wrist (*Weinman et al., 2021*).

Perturbation research at the wrist has consistently demonstrated increases in muscular co-contraction and joint stiffness in anticipation of a perturbing force (*De Serres & Milner, 1991*; *Goodin & Aminoff, 1992*; *Holmes, Tat & Keir, 2015*; *Sinkjær & Hayashi, 1989*). However, the vast majority of this research has been performed using perturbations causing wrist flexion or extension (*Holmes, Tat & Keir, 2015*; *Miscio et al., 2001*; *Weinman*

*et al., 2021*), with little consideration for radial or ulnar deviation. Investigation into the effects of radial and ulnar perturbations is needed as wrist joint dynamics are complex (*Charles & Hogan, 2010*) and primary muscles driving deviation actions can be different than flexion/extension (*Horii, An & Linscheid, 1993*). Maximal force production of the wrist toward both radial and ulnar deviation is considerably less compared to wrist flexion (*La Delfa et al., 2015*) which may lead to greater relative force needed to counteract perturbing forces. Muscle activity and co-contraction in response to radial and ulnar perturbations has been investigated during movement, with a self-selected grip force (*Forman et al., 2020b*). It was found that perturbations generating radial deviation, resulted in greater angular displacement compared to ulnar deviation. This may be due to greater strength capabilities of the wrist in resisting ulnar perturbations (*Delp, Grierson & Buchanan, 1996*; *Vanswearingen, 1983*). However, the wrist exhibits a greater moment toward ulnar deviation rather than radial deviation (*La Delfa & Potvin, 2017*).

The extensor carpi radialis, extensor digitorum communis, and brachioradialis produce significantly greater muscle activity during ulnar perturbations, while flexor digitorum superficialis, flexor carpi ulnaris, and extensor carpi ulnaris produce greater activity during radial perturbations (*Forman et al., 2020c*). However, the relationship between grip force and angular displacement cannot be commented on as grip force was neither controlled for, nor measured (*Forman et al., 2020c*). Additionally, the perturbations in this work were delivered as participants performed a dynamic tracking task. Radial and ulnar perturbations have yet to be investigated during static loading conditions. Investigation into radial and ulnar perturbations during static loading is necessary to gain insight into how the wrist is affected by perturbation direction under various grip force demands.

The purpose of this work was to investigate forearm muscle activity and wrist angular displacement during radial and ulnar wrist perturbations across various isometric hand grip demands. We hypothesized that an inverse relationship would be observed between grip force and angular wrist displacement following a perturbing force (*Holmes, Tat & Keir, 2015*; *Mogk & Keir, 2003*) and radial perturbations would result in greater angular displacement than ulnar perturbations (*Forman et al., 2020c*).

## METHODS

### Participants

This study used a robotic device to deliver wrist perturbations to human participants. University-aged individuals received a total of twenty perturbations in each direction (*i.e.*, the perturbation force caused radial/ulnar deviation) on one collection day. Post-collection data processing quantified differences in wrist angular displacement, time to peak displacement and muscle activity during radial and ulnar perturbations across four handgrip magnitudes. Twenty right-handed healthy subjects, ten males and ten females (height, 173.0 ± 10.6 cm, weight, 71.98 ± 10.2 kg, age, 22.9 ± 3.0 years) participated in this study. All participants confirmed having no history of injury to the upper limb within the last 12 months and no restriction to their upper limb range of motion. Each participant provided informed written consent prior to participation. This study was approved by the Brock University Bioscience Research Ethics Board (File # 16-263).

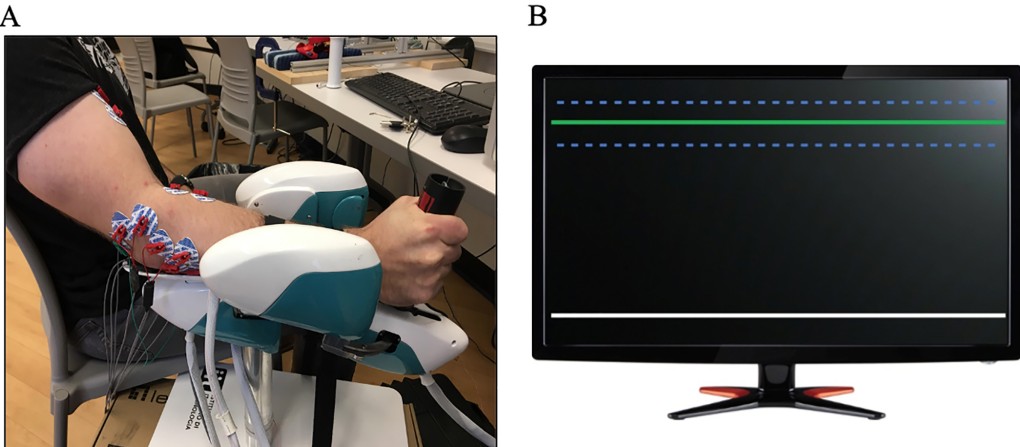

**Figure 1 Experimental protocol and participant set up.** (A) Participant completing the experimental protocol. Gripping a custom grip force handle, electromyography placed over eight forearm muscles and black Velcro straps to isolate the wrist joint. (B) The grip force visual feedback (LabView) during the protocol. Participants were instructed to grip the white line to the target depicted by the green line. Error bars were located 1.5 SD from the target and are depicted by the blue dotted lines.

## Experimental set-up

Participants were seated with their arm in a robot facing visual feedback. The robotic device was placed to the right-hand side of the body and participants rested their forearm in the device (WristBot, Genoa, Italy) and gripped a custom force sensing handle with their hand (Fig. 1A). The handle of the robot could slide to aid in unconstrained wrist rotation and this feature helped ensure a consistent starting posture for the forearm and hand across participants. Each participant grasped the handle using a power grip (with fingers and thumb wrapped around the handle) and the wrist was aligned with the centre of rotation of the robot. Shoulder and elbow joint positions were measured prior to the experiment with a goniometer (elbow flexion: $132.5 \pm 5.6°$, shoulder abduction: $34.4 \pm 5.9°$, shoulder flexion: $8.6 \pm 3.1°$, lateral rotation: $47.7 \pm 10.8°$). The WristBot is a custom-built manipulandum with a range of motion (ROM) that replicates typical human wrist motion in three degrees of freedom: flexion/extension: $\pm 62°$, radial/ulnar: $+45°/-40°$, pronation/supination: $\pm 60°$ (*Iandolo et al., 2019*; *Masia et al., 2009*). The robot uses four brushless motors to deliver torques that manipulate the wrist joint, compensate for hand mass and provide low inertia. Motors allow isolation of a targeted plane of motion, by locking the undesired degrees of freedom. Forearm pronation and supination, and wrist flexion and extension were locked during this study to control for any movement in these directions. Velcro straps were used to secure the participant's forearm to the WristBot, only allowing movement at the wrist joint.

Muscle activity was recorded with surface electromyography (EMG) over eight upper extremity muscles: flexor carpi radialis (FCR), flexor digitorum superficialis (FDS), flexor carpi ulnaris (FCU), extensor carpi radialis (ECR), extensor digitorum communis (EDC), extensor carpi ulnaris (ECU), brachioradialis (BR) and biceps brachii (BB). Disposable paired monopolar Ag-AgCl electrodes (MediTrace 130; Kendall, Mansfield, MA, USA)

were placed over the muscle belly of each muscle, in-line with muscle fiber orientation, and with an inter-electrode distance of 2.5 cm. Electrode placement was determined with palpation following established guidelines (*Perotto & Delagi, 2005*) and the recording sites were shaved and scrubbed with alcohol prior to placement. EMG signals were band-pass filtered (10–1,000 Hz), differentially amplified (CMRR > 115 dB at 60 Hz; input impedance, ~10 GΩ; AMT-8, Bortec Biomedical Ltd, Calgary, AB, Canada), sampled at 2,000 Hz (USB-6229 BNC; National Instruments, Austin, TX, USA) and synched with robot kinematics. To normalize EMG signals, 2 maximum voluntary contractions (MVC) were performed for each muscle using muscle specific contractions for the eight muscles. Participants were manually resisted by the researcher following the maximum contraction protocol outline by (*Forman et al., 2020a*). Each MVC was held for 5 s, and a min of 30 s rest was given in between contractions.

## Experimental protocol

Maximum grip force was measured using a custom grip force handle equipped with a force transducer. Participants performed two maximum voluntary grip trials using the force sensing handle. If these two grip force trials differed by more than 5%, a third trial was performed (*Antony & Keir, 2010*; *Au & Keir, 2007*). The greatest peak force from the maximum grip trials was used to normalize the target grip forces for experimental conditions. During experimental conditions, participants were instructed to grip the handle of the robotic device to a specified target grip magnitude. Real time grip force was displayed on a monitor directly in front of the participant and the target grip magnitude was indicated by a horizontal line (LabView 2016; National Instruments, Austin, TX, USA). Additional horizontal lines represented error bars positioned ±1.5% MVC around the target grip (see Fig. 1B).

Participants became familiarized with gripping to the visual target as well as the force and timing of the perturbation. Once familiar with the robotic device and experimental set-up, participants performed gripping tasks with targets normalized to 5/20/50/80% of their maximum grip strength (% MVG). Every 7 s, the participant received a perturbation in the radial or ulnar direction, that caused rotation of the hand at the wrist joint. Participants were made aware of the perturbation timing and were prompted by the researchers to grip to the target force on the monitor prior to receiving a perturbation. Perturbation timing was made known to the participant to minimize muscular fatigue. Participants gripped to the target force prior and during the perturbation and relaxed until the onset of the next perturbation. There were five perturbation trials for each direction (radial deviation (RD) and ulnar deviation (UD)) at each of the four grip force conditions. Perturbation direction was known to participants and grip force conditions were randomized.

In a neutral wrist posture, the robot delivered perturbations at a force of 22.3 N, resulting in an end effector wrist torque of 1.78 Nm over 100 ms. This force was chosen based on previous literature that delivered wrist perturbations (*Forman et al., 2020c*; *Holmes, Tat & Keir, 2015*). Participants were instructed to focus on maintaining the required grip force rather than hand position and they were not instructed to resist the

device. A perturbation warning was given 2 s before the perturbation which allowed adequate time for the participant to reach the required grip force target before the perturbation (without a long grip hold that may cause fatigue). The perturbations occurred every 7 s and was made aware to the participant. Rest was given between trials and conditions based on the target grip magnitude (5% MVG: 30 s, 20% MVG: 60 s, 50% MVG: 90 s, 80% MVG: 120 s), to minimize fatigue. Maximal grip force was also measured post-experiment to assess fatigue.

## Data analysis

All EMG and kinematic data were analyzed offline using MATLAB 2015b (MathWorks Inc., Natick, MA, USA). A quiet EMG trial was collected prior to the experimental conditions, and the bias was removed from each EMG channel. The EMG data (both maximal contraction and experimental trials) were linear enveloped (rectified and low pass Butterworth filtered using a dual pass, $2^{nd}$ order and 3 Hz cut-off). Considering stiffness is dependent on muscle force, our signal processing may reflect an EMG-muscle force relationship (*Winter, 2009*). This processing would allow for future joint rotational stiffness evaluations (*Potvin & Brown, 2005*; *Holmes, Tat & Keir, 2015*). EMG from experimental trials were normalized to the greatest muscle specific maximal voluntary contraction (% MVC). Average muscle activity was measured at three time points (*Pruszynski & Scott, 2012*); (1) Baseline (PRE) 0–15 ms prior to perturbation, (2) POST1, 20–50 ms after perturbation and (3) POST2, 50–100 ms after perturbation. Angular displacement (AD) was computed and stored by the robot and measured the displacement of the WristBot handle from the participants starting wrist position to peak displacement after the perturbation. Time to peak displacement (TPD) was recorded as the time to the maximum angular displacement following the onset of the perturbation. Wrist angular velocity was calculated from the mean AD (°) and TPD (ms) across all five trials for each of the four grip force conditions and perturbation directions. In addition, percent changes of angular velocity were calculated for each of the grip forces and directions. Kinematics were measured by the robot and sampled at 100 Hz. AD, TPD and muscle activity (EMG) was measured pre-and post-perturbation to evaluate the influence of grip force on wrist joint kinematics.

## Statistical analysis

Statistical analysis focused on comparing the response of the kinematic and EMG data to AD following a perturbation. A paired t-test was conducted to compare mean grip force prior to radial or ulnar perturbations for each grip intensity. A 2 (perturbation direction) × 4 (grip force) factorial ANOVAs with repeated measures was used to assess AD, TPD and angular velocity. A 2 (perturbation direction) × 3 (time) × 4 (grip force) factorial ANOVA with repeated measures was used to assess EMG activity across perturbation direction (RD/UD), time point (PRE, POST1, POST2), and grip magnitude (5/20/50/80% maximum). All outcome measures were averaged across the five repetitions for each experimental condition, for each participant. Assumptions of normality, homogeneity of variance and sphericity were all met ($p > 0.05$). An alpha level of $p < 0.05$

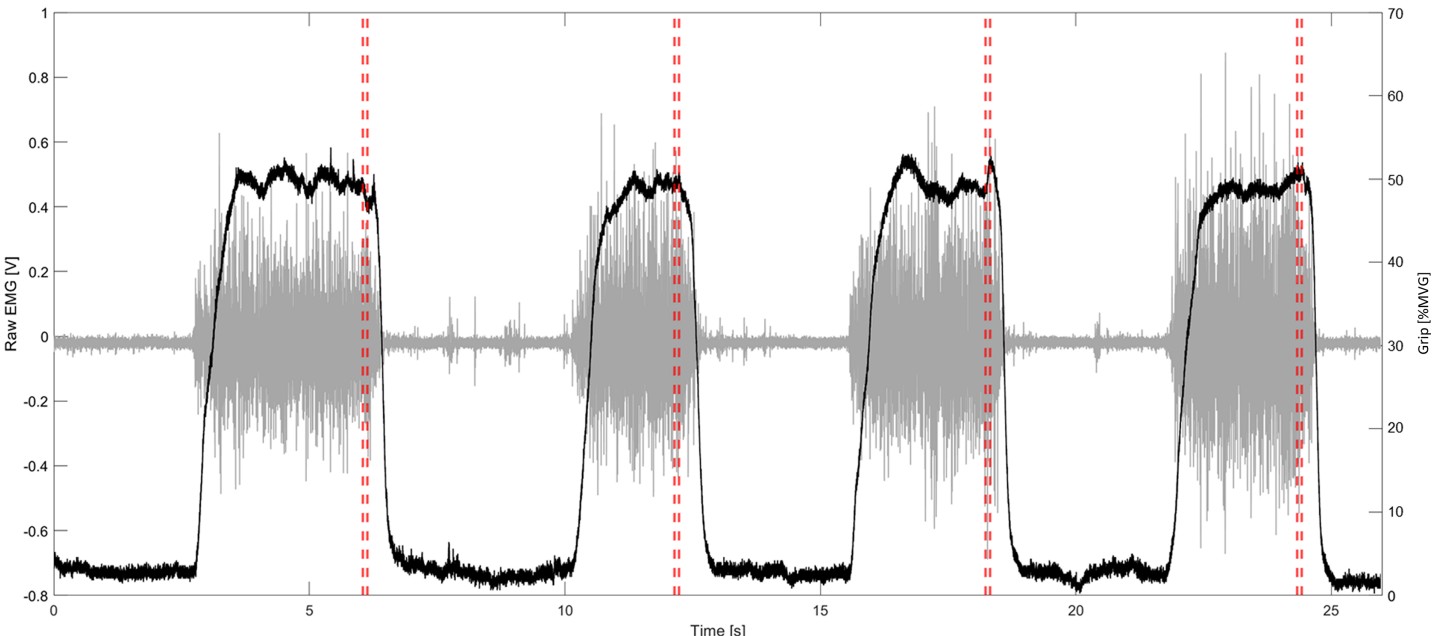

**Figure 2 Representative data from participant.** Representative data of one trial (four perturbations, one in each direction). Grey is raw EMG of a representative muscle; Black is the grip force normalized to maximum grip; vertical red lines indicate the start of the perturbation.

was used, and a Bonferroni post-hoc test was chosen to assess interaction effects. All data are presented as mean ± standard error.

# RESULTS

## Grip force

No significant differences ($p = 0.82$) were found between grip and direction as mean grip forces were consistently held in RD and UD perturbations equally. No changes in maximum grip force were found pre ($292.8 \pm 76.1$ N) and post experiment ($301.5 \pm 80$ N), indicating that muscle fatigue did not influence the results of the study ($p = 0.69$). Figure 2 demonstrates raw grip force and muscle activity for a representative participant, with perturbation onset indicated.

## Wrist kinematics

There was a main effect of grip force on AD ($p \leq 0.001$, $F_{(3, 19)} = 631.06$, $\eta^2 = 1.00$), a Bonferroni post-hoc analysis confirmed significantly less angular displacement as grip force increased ($p = 0.001$) (Fig. 3). A main effect of direction was also found ($p \leq 0.001$, $F_{(3, 19)} = 1256.19$, $\eta^2 = 0.99$). There was higher mean angular displacement in RD compared to UD (RD: $17.31 \pm 0.59°$, UD: $10.43 \pm 0.47°$). No significant interaction was found between grip force and perturbation direction for peak AD ($p = 0.75$, $F_{(3,19)} = 0.41$, $\eta^2 = 0.21$) (Fig. 3).

A significant main effect of grip force on TPD was found, indicating that TPD significantly decreased as grip force increased, regardless of perturbation direction (Fig. 4). A main effect of direction was found ($p \leq 0.001$, $F_{(3,19)} = 2035.33$, $\eta^2 = 0.99$) and there was a

Peerj

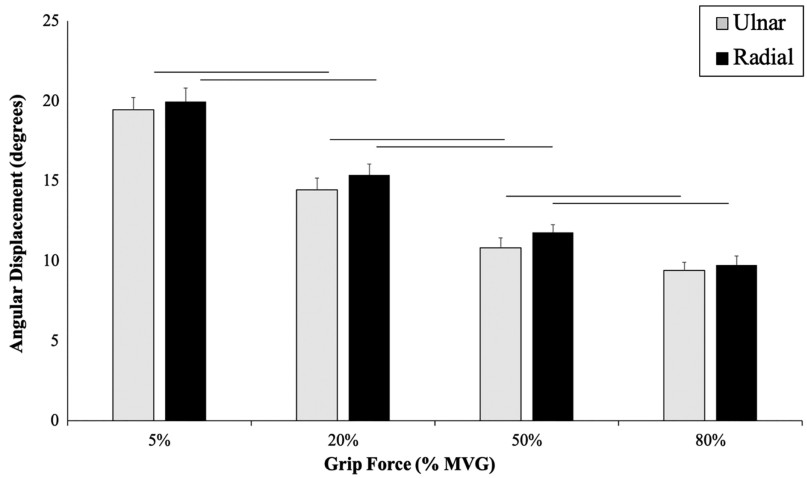

**Figure 3 Angular wrist displacement measured during each grip condition.** Angular displacement (degrees) across four grip conditions in UD (grey) and RD (black). Main effects of AD, grip force and direction are presented. There was less angular displacement as grip force increased ($p \leq 0.001$) and a higher mean angular displacement in RD compared to UD ($p \leq 0.001$).

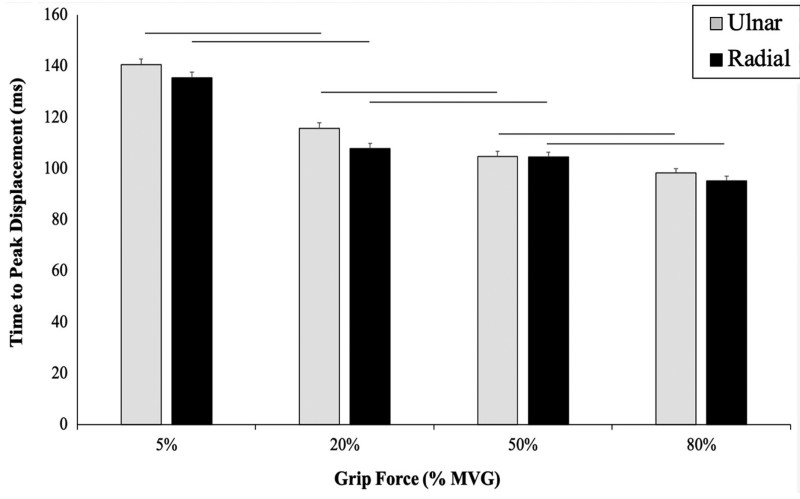

**Figure 4 Time to peak displacement measured for each grip condition.** Time to peak displacement across 4 grip conditions in UD (grey) and RD (black). TPD displayed as (ms) along the y-axis. TPD significantly decreased as grip force increased ($p \leq 0.001$). A main effect of direction ($p \leq 0.001$) as there was a significantly slower TPD in RD. No significant interaction was found between TPD (ms) and grip force ($p = 0.31$).

significantly slower TPD in RD (RD: 129.06 ± 1.77 ms, UD: 100.65 ± 1.67 ms). No significant interaction was found between TPD (ms) and grip force ($p = 0.31$, $F_{(3,19)} = 3.170$, $\eta^2 = 0.71$) (Fig. 4).

Percent change scores for angular velocity were calculated between two consecutive measurements: 5–20%, 20–50%, 50–80% MVG (Fig. 5). No significant interactions were found for angular velocity ($p = 0.84$) and no significant main effects of grip and direction (p's = 0.25).

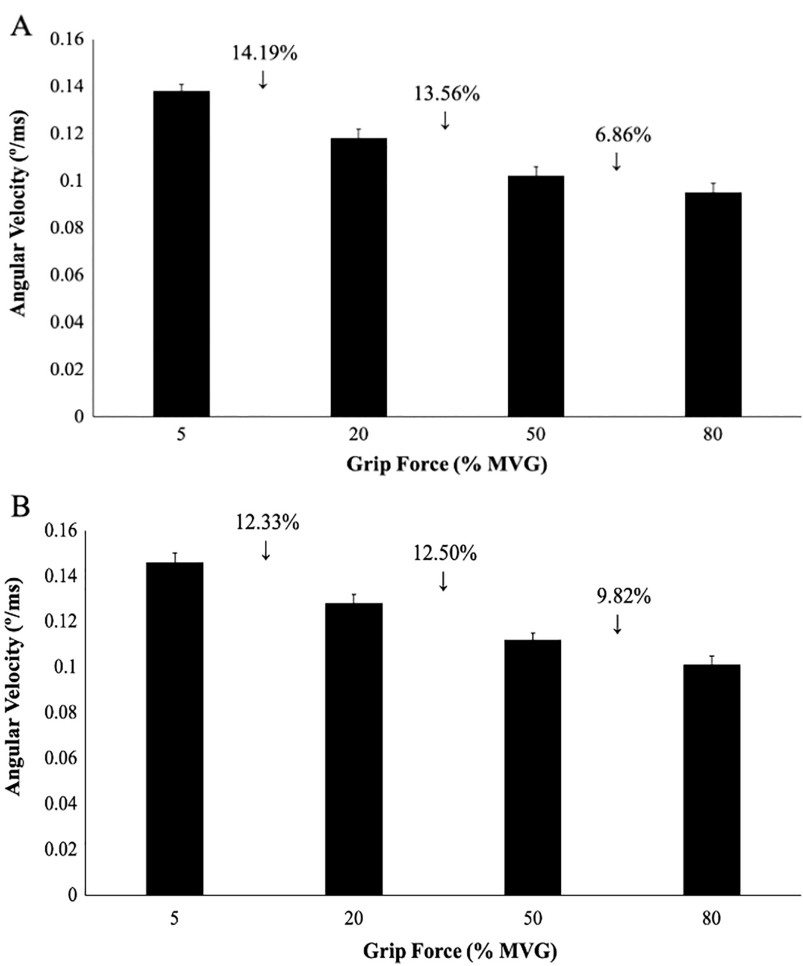

**Figure 5 Angular wrist velocity for radial and ulnar perturbation directions.** Group averages for angular velocity (degrees/ms) and percent change scores. (A) Angular velocity in RD, (B) Angular velocity in UD. No significant interactions or main effects were found for angular velocity despite the percent change between each grip condition.     

## EMG and muscle activation responses

There was a significant main effect of direction for ECR, EDC and BR ($p = 0.02$, $F_{(1, 19)} = 5.69$, $\eta^2 = 0.12$). A Bonferroni post-hoc analysis revealed significantly larger mean muscle activation in UD as compared to RD for all grip conditions. No significant effect of time was found across any muscles (Table 1). Extensor muscles (RD: 23.27% ± 16.45% MVC, UD: 24.00% ± 17.04% MVC) produced greater muscle activation compared to flexor muscles (RD: 17.19% ± 15.59% MVC, UD: 16.25% ± 15.40% MVC) across all grip forces.

No significant interactions were found between AD, TPD and grip conditions (Table 1). A significant main effect of grip force magnitude was observed across all muscles. There was an increase in muscle activation with increasing grip forces across all muscles ($p \leq 0.001$, $F_{(3, 19)} = 774.19$, $\eta^2 = 0.84$) excluding BB which showed no differences in muscle activity throughout ($p = 0.72$) (Figs. 6 and 7).

**Table 1 Significant main effects and interactions for muscle activity across all muscles. Statistically significant, $p < 0.05^*$.**

Muscle activity $p$-values

| Muscle | Direction | Time | Grip | Direction × time | Direction × grip | Time × grip | Direction × time × grip |
|---|---|---|---|---|---|---|---|
| FCR | 0.203 | 0.855 | <0.001* | 0.897 | 0.855 | 0.897 | 0.997 |
| FDS | 0.209 | 0.947 | <0.001* | 0.541 | 0.987 | 0.942 | 0.996 |
| FCU | 0.214 | 0.950 | <0.001* | 0.565 | 0.472 | 0.709 | 0.999 |
| ECR | <0.001* | 0.445 | <0.001* | 1.264 | 5.053 | 0.809 | 0.989 |
| EDC | <0.001* | 0.763 | <0.001* | 0.405 | 0.531 | 0.870 | 1.000 |
| ECU | 0.323 | 0.096 | <0.001* | 0.596 | 0.826 | 0.411 | 0.994 |
| BR | 0.006* | 0.576 | <0.001* | 0.801 | 0.393 | 0.999 | 1.000 |
| BB | 0.797 | 0.916 | 0.722 | 0.943 | 0.992 | 0.995 | 0.999 |

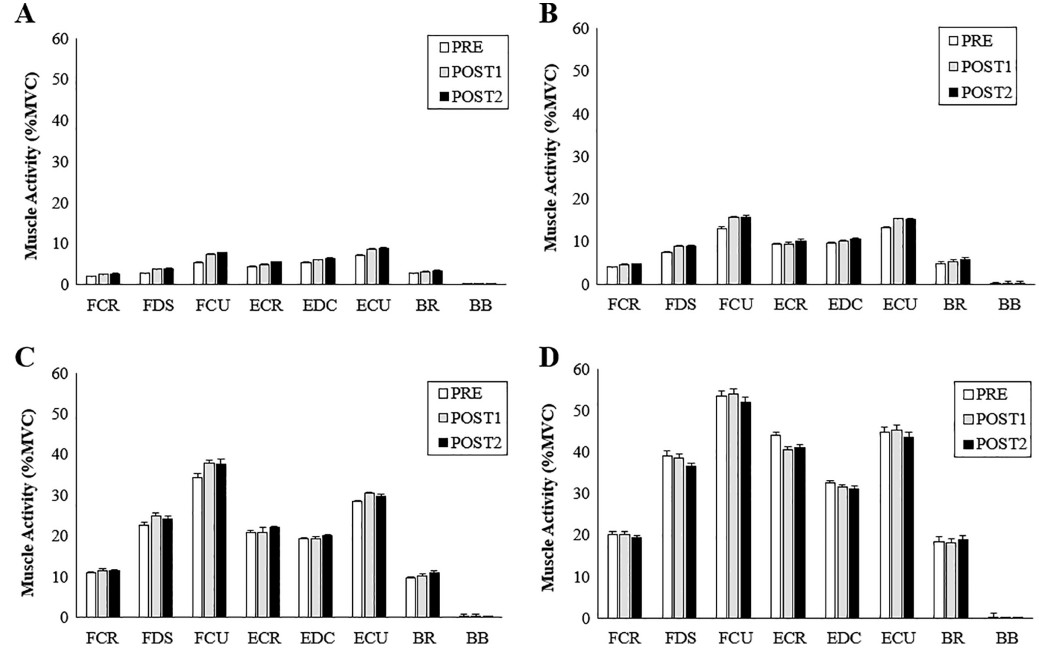

**Figure 6 Muscle activity during radial perturbations for each grip condition.** Group averages for muscle activity (%MVC) for RD. (A) 5% grip force, (B) 20% grip force, (C) 50% grip force and, (D) 80% grip force.

## DISCUSSION

The purpose of this study was to determine how radial and ulnar perturbations affect the wrist joint musculature during isometric gripping and to explore the relationship between grip force magnitude and angular wrist kinematics. Results support our first hypothesis where a greater grip force resulted in a smaller angular wrist displacement following a perturbing force in both radial and ulnar deviation. Indeed, there was a significant decrease in angular displacement with increasing grip force and this corresponds with an increase in muscle activity, suggesting that joint stiffness increased with an increase in muscle activity (*Franklin & Milner, 2003*). Although wrist angular displacement significantly

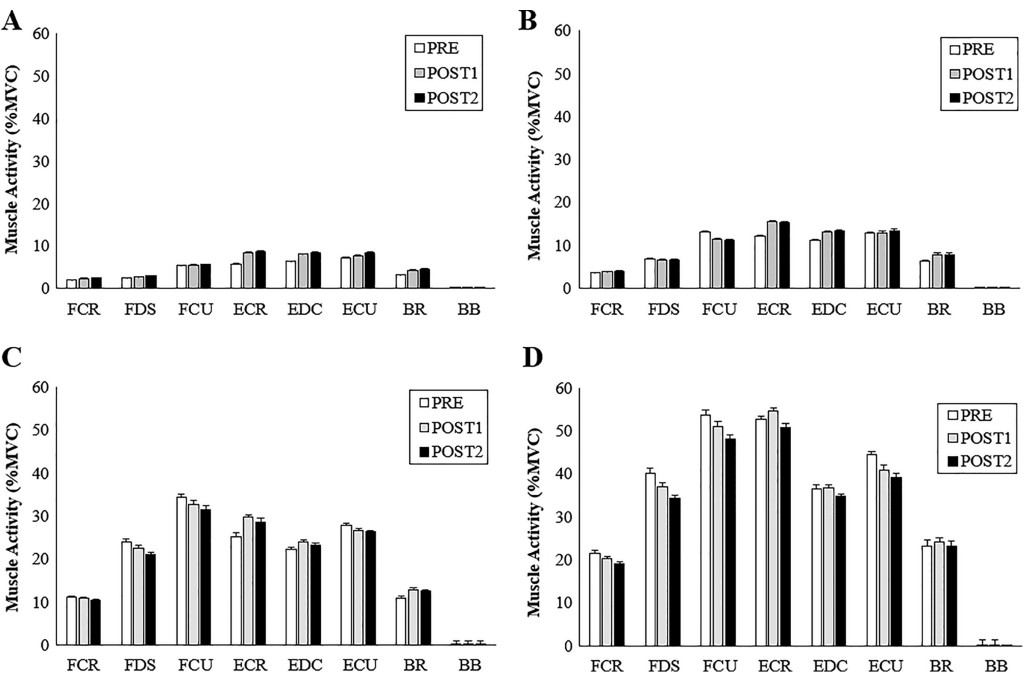

**Figure 7 Muscle activity during ulnar perturbations for each grip condition.** Group averages for muscle activity (%MVC) for UD. (A) 5% grip force, (B) 20% grip force, (C) 50% grip force and (D) 80% grip force.

decreased by 2.02° from the 50% to 80% grip conditions, this difference was 4.18° between 20% to 50% MVG, considering both directions. Similarly, angular velocity was 13.56% slower when gripping from 20% to 50% MVG in radial deviation and 6.85% slower when gripping from 50% to 80% MVG. Thus, increasing grip demands from 50% to 80% MVG may not provide additional benefit for minimizing joint rotation.

Results of the study also support hypothesis 2; radial perturbations had a significantly greater mean angular displacement (17.31 ± 0.59°) compared to ulnar perturbations (10.43 ± 0.47°) across all grip conditions. This finding may be explained by the strength capabilities of the forearm muscles. ECR and EDC produced greater activity in UD as compared to RD for all grip conditions. This is supported by findings of greater angular displacement following a perturbation that moves the wrist into radial deviation compared to ulnar deviation (*Forman et al., 2020c*). The forearm extensors contribute substantially to wrist joint stability (*Holmes, Tat & Keir, 2015*; *Holzbaur, Murray & Delp, 2005*), due to a combination of muscle activity and geometric (*i.e.*, line of action and moment arm) advantages as compared to the flexors (*Gonzalez, Buchanan & Delp, 1997*). Further, ECR activity may be prioritized when opposing ulnar perturbations due to a more direct muscle line of action (*Bawa et al., 2000*; *Horii, An & Linscheid, 1993*). Our noted muscle activity differences could contribute to the greater angular displacement exhibited during radial perturbations compared to ulnar perturbations. It should be noted that our experimental set up required participants to grasp a handle using a power grip. This requirement could favour radial deviation. Future work may want to consider the roles of

extensor pollicis longus and extensor pollicis brevis, which can contribute to wrist deviation and extension, *via* musculoskeletal modelling.

Deviation of the wrist from a neutral position, particularly while generating high wrist or grip force requirements can pose as a risk factor for musculoskeletal disorders. Experiencing perturbations in the workplace (either from tool forces or unexpected sudden loading) can cause rapid muscle stretch and eccentric forces, which may, over time be a contributing factor for injury development. Two adaptive behaviours can occur in response to a sudden perturbation, it can be either proactive–an anticipatory response, or reactive–a compensatory response to a sudden disturbance (*Stuphorn & Emeric, 2012*). In this work, no significance was found across muscles for any of the measured time points (PRE, POST1, and POST2). Background muscle activity is thought to influence post-perturbation and voluntary responses differently (*Pruszynski et al., 2009*). In our work, there appeared to be a transition from a reactive to proactive muscle recruitment strategy as grip force increased. An emerging trend demonstrates more of a reactive response associated with lower grip forces (5% MVG and 20% MVG) as POST2 had greater muscle activity compared to higher grip conditions. Whereas a proactive response is visible with the higher grip forces (50% MVG and 80% MVG) as muscles are more active during the pre-perturbation phase, generating wrist stiffness and resulting in a smaller post response. These trends are demonstrated following both radial and ulnar perturbations. These responses could have been a result of the participants knowledge of perturbation timing (*Forgaard et al., 2016*) thus, creating a proactive response before the perturbation occurred with higher grip forces. These results are similar to work done by *Holmes & Keir (2014)*, who found that known perturbations result in a voluntary muscle response to prematurely stiffen the joint and unknown perturbations result in a greater post or involuntary response.

Another possibility as to why POST2 responses had a larger contribution in lower grip force conditions could be due to the focus of attention, velocity, and muscle activation at the different grip intensities. At lower grip forces, the participants required minimal effort and little muscle activity to obtain 5% and 20% of their maximum grip force and were probably less focused on the perturbations, which may have resulted in a larger reactive response. In the higher grip force scenarios participants may have focused more of their attention on the higher effort gripping task directly before the perturbation; by focusing more on the timing of the perturbation component they may have created an increase in voluntary muscle contraction that prematurely stiffened the wrist. Previous work using an applied pre-loading wrist torque producing approximately 10% of maximum muscle activity to investigate short latency (SLR) and long latency reflexes (LLR) could be considered similar to our low grip force condition (*Miscio et al., 2001*). *Miscio et al. (2001)* found smaller LLR amplitudes for muscles that shorten during perturbation as compared to muscles that lengthen (flexion/extension direction). However, *Weinman et al. (2021)* found significant modulation of LLR in shortened muscles, and this was influenced by perturbation velocity, duration, background torque and task instruction and represents a physiological decoupling of various muscle responses. Our trends speculate that there are differing neuromuscular patterns associated with changes in grip force (pre-loading). Finally, a more general effect could be due to

higher level motor control mechanisms that intervene in the planning of voluntary contraction (*Morasso et al., 2014*).

## LIMITATIONS

This study only used University-aged participants and thus, the results of this work may be difficult to generalize across all populations or age groups. It is important to note that crosstalk is always a considerable factor when using surface EMG of the forearm. However, careful placement of the electrodes (*Perotto & Delagi, 2005*) and skin preparation guidelines were followed to minimize this risk. To control for the many variables that could play a role in the experiment, we used an innovative, quantitative, and repeatable robot-based perturbation delivery. There was inherent bias due to forearm position. Participants were positioned in the robotic device in a neutral wrist posture (defined as mid-pronation) and were instructed to grasp the handle using a power grip. Extensor pollicis longus and extensor pollicis brevis, which can contribute to wrist deviation and extension, were not measured given potential limitations with surface EMG from these muscles. Due to the wrist posture associated with grasping the handle, these muscles could be active during radial deviation and wrist extension. Future work should investigate wrist stiffness during perturbations from different starting postures and consider contributions from the hand/ finger musculature. Future work should investigate the onset of muscle fatigue (*Mugnosso et al., 2018*), the assessment of pain (*Albanese et al., 2019*), changes in wrist joint stiffness (*Falzarano et al., 2020*) and the role of proprioception (*Albanese et al., 2021*).

## CONCLUSIONS

This work sought to understand the effects of grip force (as a determinant of changing wrist joint stiffness) in response to sudden perturbing forces in radial and ulnar deviation. It is important to understand the point of diminishing returns in tasks that require gripping. Results of this study show a significant decrease in wrist angular displacement and angular velocity as grip force increases, with a slight plateau between 50–80% maximum grip. An increase in angular displacement and slower angular velocity following radial perturbations suggest that sudden radial displacement of the wrist joint may need to be investigated further. Lastly, there was a noticeable but non-significant trend of reactive muscular responses in the lower grip forces compared to a proactive response in higher grip forces, indicating potential differences in neuromuscular patterns associated with altered grip force conditions. Further work investigating co-contraction and joint stiffness in response to radial/ulnar perturbations are necessary to contribute to the understanding of the relationship between grip force and wrist joint stiffness.

### Funding

This work was funded by an NSERC Discovery Grant and the Canada Research Chairs Program. The funders had no role in study design, data collection and analysis, decision to publish, or preparation of the manuscript.

## Grant Disclosures

The following grant information was disclosed by the authors:

NSERC Discovery Grant.

Canada Research Chairs Program.

## Competing Interests

Michael W.R. Holmes is an Academic Editor for PeerJ.

## Author Contributions

- Kailynn Mannella conceived and designed the experiments, performed the experiments, analyzed the data, prepared figures and/or tables, authored or reviewed drafts of the paper, and approved the final draft.
- Garrick N. Forman conceived and designed the experiments, performed the experiments, analyzed the data, prepared figures and/or tables, authored or reviewed drafts of the paper, and approved the final draft.
- Maddalena Mugnosso conceived and designed the experiments, performed the experiments, analyzed the data, prepared figures and/or tables, authored or reviewed drafts of the paper, and approved the final draft.
- Jacopo Zenzeri conceived and designed the experiments, analyzed the data, authored or reviewed drafts of the paper, and approved the final draft.
- Michael W.R. Holmes conceived and designed the experiments, analyzed the data, authored or reviewed drafts of the paper, and approved the final draft.

## Human Ethics

The following information was supplied relating to ethical approvals (*i.e.*, approving body and any reference numbers):

Brock University Research Ethics Board granted approval of this work (16-263).

## Data Availability

The muscle activity and kinematics data are available in the Supplemental File.

## Supplemental Information

Supplemental information for this article can be found online at http://dx.doi.org/10.7717/peerj.13495#supplemental-information.

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
