# Peer review of "The effects of isometric hand grip force on wrist kinematics and forearm muscle activity during radial and ulnar wrist joint perturbations"

_PeerJ, doi:10.7717/peerj.13495_

## Round 0.1 · original submission · Major Revisions

Please write a point by point rebuttal letter, explaining how you addressed the reviewer comments.

Although not strictly required, I would recommend that you provide your original data with the manuscript. Experience has shown that this can increase the impact and citations of a publication. See:
https://peerj.com/about/policies-and-procedures/

·

Basic reporting

This paper describes a study on the effects of hand grip force on the kinematic and EMG responses to perturbations of wrist posture into ulnar and radial deviation. Not surprisingly the results show decreasing wrist angular displacement and displacement velocity and higher EMG amplitudes with increasing grip force.
I miss a clear rationale for the study. The introduction sketches a practical rationale related to musculoskeletal loading and injury, which is not clearly related to the experimental tasks and conditions. The authors state that continuous muscular activation above 1-5% of maximal could be a cause of injury. First of all, I would like to comment that the reference provided does not provide solid evidence to support this, but more importantly I do not see how this is related to the effect of perturbations and their dependence on grip force. The authors state that understanding the effect of perturbations on task performance and injury risk is critical, but it is unclear why and neither of these outcomes are investigated. The aim should most likely be limited to adding incremental knowledge on wrist joint stiffness in line with predictions that can be made on theoretical and empirical grounds. This would allow shortening the introduction substantially and make it more to the point.

Specific comments
Line 46: “are increasing” should be “is increasing”
Line 65: “surrogate of wrist stiffness” Co-contraction is a determinant of stiffness, but it seems odd to see it as a surrogate.
Line 82: 84: This does not follow logically from the text above. In addition, how does this experiment help us to “understand optimal joint stiffness for task performance and injury mitigation”?
Line 86: Co-contraction can be used to increase joint stiffness in anticipation of a perturbation. Increasing stiffness in response to a perturbation does not make sense.
Results section: It would be good to sow some of the raw kinematic and EMG data.
Line 278: What is “meaningful”?
Line 283: “A decrease in angular displacement due to greater muscle activity when resisting an ulnar perturbation could suggest that the extensor muscles produce more force in all grip conditions as compared to flexor muscles”. This is unclear to me.
Lines 288-290: “In addition, the greater angular displacement following radial perturbations compared to ulnar perturbations is in line with greater cocontraction of FCU/ECU during a radial perturbation compared to FCR/ECR.” Again, this is unclear to me.
Line 300: Why would a larger displacement entail a higher injury risk. This can be the case but the opposite could also be true.
Lines 320 a.f. Why would a lower grip force reduce attention to the perturbation? I would expect the opposite from the perspective of resource competition.
Line 352: How is continuous exertion related to perturbation responses?
Lines 359-360: If I’m not mistaken there was no significant interaction between time window and grip force so this conclusion would be flawed.

Experimental design

Overall, the experiment appears carefully performed and it is for the major part clearly reported. I do have some concerns with respect to the experimental design, data analysis and interpretation which are given below.
Lines 149-151: The biceps brachii and brachioradialis muscles are not commonly seen as forearm muscles and neither of these spans the wrist joint. It is unclear to me why these muscles were included.
Line 160: Was there only a single repetition for the MVC measurements for each muscle? This would be a limitation both regarding the statistical precision and validity of the normalized EMG amplitudes.
Line 166: Similarly, the use of just two repetitions to determine maximal grip force is a limitation.
Line 181 a.f. Were participants instructed to resist the perturbation or to let it happen? How long before the perturbation did they receive a warning? Why was perturbation direction revealed?
Lines 196-211: The order of processing steps is unclear. It would be to describe data acquisition and processing steps in the order in which they were performed. This also holds for the kinematic data. I assume that EMG data were rectified before low-pass filtering. Importantly, the 3Hz low-pass filter is at odds with the time windows to analyze pre-activation, and short and long-latency reflexes. If one wants to separate these time windows a high temporal resolution so high cut-off frequency is indicated.

Validity of the findings

The EMG findings have limited precision and validity in view of the concerns raised above.

Additional comments

n.a.

Reviewer 2 ·

Basic reporting

Well written article, content is logically organized, hypotheses stated, and literature referenced appropriately.

Experimental design

Well conceived experimental design, nice example of attempt to measure EMG activity from a large set of forearm muscles.

Validity of the findings

The findings are mostly valid, however there is one statement about "increasing grip force beyond 50% of maximum does not substantially reduce angular displacement about the wrist joint." which is actually contradicted by the data presented by the authors.

Additional comments

This is an interesting and well written paper about the effects of grip force on wrist kinematics and activity of forearm muscles during perturbations of the wrist joint in radial-ulnar deviation.

I only have one major concern, which is the point of "diminishing returns" in the effect of grip force on the reduction in the amplitude and timing of the perturbations. The observations provided by the authors here do not seem to be supported by their data nor analysis (Fig. 2, Fig. 3). I would suggest the authors to remove such a statement from the abstract, and probably rephrase the paragraph in lines 274-279 (and 354) or, in alternative, provide quantitative support for this claim.

Minor comments:
line 124 - please indicate if the height/weight/age figures are mean +- st. deviation, s.e.m., or c.i.?
line 155 - please fix reference to textbook
line 157 - is the input impedance 10 GΩ?
line 168 - how was the custom instrumented handle calibrated to detect a measurement in newtons? I understand that all magnitudes are referred in terms of the MVC, but the authors hint to error bars extending +- 1.5 N around the target grip force value
line 207 - how are decent changes defined? is it the percent change between two consecutive measurements? which order, and normalized how?
lines 214-220 - is the statistical analysis run on the average measurement collected from each individual per each condition, or including all repetitions? the dataset seem to only use one measurement per subject per condition. please specify what method is used and why. also - please specify if the mixed model is estimating a subject-specific intercept as random effect (only), or more subject specific terms
line 231 and below - please specify which outcome is used at the beginning of each paragraph, so it's clear if the effect - say - of grip force is on AD, TPD, or others
line 244 - unclear what a "great" percent change score is

---

## Round 0.2 · Minor Revisions

We are very close to accepting your manuscript for publication. Please submit a revision, and make sure you address the reviewer comments in your rebuttal and revision. If I am satisfied with the revision, I can accept it without further review. If I am not sure, I will send it back to one or both reviewers.

Reviewer 2 ·

Basic reporting

Well written article, content is logically organized, hypotheses stated, and literature referenced appropriately.

Experimental design

Well conceived experimental design, nice example of attempt to measure EMG activity from a large set of forearm muscles.

Validity of the findings

Findings are valid

Additional comments

Just some minor comments to improve the clarity.

Abstract: you say that there was an increase in muscle activity with higher grip forces across all muscles.
Please specify whether this applies pre-, post-perturbation, and at which time instants post-perturbations (previous sentence in the abstract introduces quantification of EMG at multiple timepoints).

Methods: How different did the grip force trial need to be to warrant collection of a third trial? I would assume that two repeated measurements of grip force are *always* different provided that the sensor has sufficient resolution.

Reviewer 3 ·

Basic reporting

The manuscript is clearly written. The changes based on previous reviewer comments were well done generally.

Experimental design

Biases due to forearm position should be addressed - were there inherent biases due to setup? Ie. Mid-prone (thumb up) may bias radial deviators. Wrist posture and RU deviation may limit stiffness via different rOMs.

Validity of the findings

There was a question of using a 3Hz linear envelope. The authors chose not to re-analyze the data but simply change the name of the variables. Renaming variables does not alter the implicit construct of the meaning of the timing. If solely discussing these timings with respect to reflex loops and long vs short latency windows, then this is a poor choice as the 3 Hz LPF is not approriate. However, I believe the authors' effort is to address stiffness which would be dependent on muscle force - which in turn is related to a 3Hz (or actually lower Fc). My suggestion is to embrace the timing of muscle force rather than reflex timing.

The big concern for me in the discussion of finding is that the thumb muscles were not addressed - in particular EPL and EPB - which act often as wrist deviators and extensors - I believe particularly so in this posture. At a minimum, this should be addressed.

The added raw data figure should have actual converted units rather than just volts. Using volts does not allow the reader to appreciate your force consistency - that is, how much variability is present.

---

## Round 0.3 · accepted · Accept

Thank you for the final revision. The manuscript is now acceptable for publication.